# Proposed Framework for Conducting Clinically Relevant Translational Biomarker Research for the Diagnosis, Prognosis and Management of Sepsis

**DOI:** 10.3390/diagnostics14030300

**Published:** 2024-01-30

**Authors:** François Ventura, Gilbert Greub, W. Conrad Liles, Shevin T. Jacob

**Affiliations:** 1Division of Anesthesiology, Geneva University Hospitals, Rue Gabrielle-Perret-Gentil 4, CH-1211 Geneva, Switzerland; 2Intensive Care Unit, Hirslanden Clinique des Grangettes, Chemin des Grangettes 7, CH-1224 Chêne-Bougeries, Switzerland; 3Medical Microbiology, Centre Hospitalier Universitaire Vaudois, Rue du Bugnon 46, CH-1011 Lausanne, Switzerland; gilbert.greub@chuv.ch; 4Institute of Microbiology, University of Lausanne, Rue du Bugnon 48, CH-1005 Lausanne, Switzerland; 5Sepsis Center of Research Excellence, University of Washington, Seattle, WA 98195-6420, USA; wcliles@medicine.washington.edu; 6Departments of Medicine, University of Washington, Seattle, WA 98195-6420, USA; 7Department of Clinical Sciences, Liverpool School of Tropical Medicine, Pembroke Place, Liverpool L3 5QA, UK; shevin.jacob@lstmed.ac.uk; 8Walimu, Kampala P.O. Box 9924, Uganda

**Keywords:** infection, systemic inflammation response, sepsis, septic shock, organ failure, organ dysfunction, biomarker, antibiotics, antimicrobial resistance

## Abstract

Although the diagnosis of sepsis requires the identification of the three components of infection, a systemic inflammation response, and organ dysfunction, there is currently no consensus on gold-standard criteria. There are however suggested tools and tests, which have been proposed in international guidelines, including those produced by the Surviving Sepsis Campaign. Biomarkers play an important role in these tools and tests, and numerous heterogeneous studies have been performed to evaluate their respective clinical utility. Our review of the current practice shows that no biomarkers of infection, systemic inflammation response, organ dysfunction and sepsis are currently specifically recommended, which is probably due to the lack of standardization of studies. We therefore propose to define a framework for conducting clinically relevant translational biomarker research and seek to establish ideal criteria that can be applied to an infection, systemic inflammation response, organ dysfunction and sepsis biomarkers, which can enable early screening of sepsis, diagnosis of sepsis at the time of clinical suspicion and monitoring of sepsis treatment efficacy.

## 1. Introduction

Infectious diseases are a major public health problem throughout the world, and infections can be complicated by sepsis, an organ dysfunction caused by a dysregulated response to infection [1]. Septic shock is a subset of sepsis, in which circulatory collapse and metabolic dysfunction are associated with end-organ failure and high mortality. Worldwide, an estimated 48.9 million cases of sepsis occur each year, resulting in 11 million deaths [2]. Within the Global Burden of Disease, which records hierarchical levels of causes of health loss, the proportion of global deaths from sepsis (which is associated with 33 clinically significant bacterial pathogens) establishes it as the second leading level 3 cause of worldwide mortality [3,4].

The diagnosis and management of sepsis are based on criteria and algorithms published in international guidelines [5]. The definition of sepsis has evolved over time. In 1992, it was first defined, by consensus, to be a suspected infection associated with a systemic inflammatory response syndrome (SIRS) [6]; categories of sepsis, severe sepsis and septic shock were then delineated to define sepsis along a spectrum that corresponds to the host response to infection. A 2001 definition retained the unchanged 1992 definition, but expanded the list of signs and symptoms of sepsis to reflect clinical bedside experience and incorporate general, inflammatory (e.g., increased plasma C-reactive protein or increased plasma procalcitonin), hemodynamic, and tissue perfusion (e.g., hyperlactatemia) parameters [7]. Notably, although one objective of the 2001 meeting was to incorporate biomarkers into a new sepsis definition, it was held at the time that the development of a biomarker-based definition was premature. In 2016, a third set of definitions of sepsis was proposed, which defined it as organ dysfunction caused by a dysregulated host immune response (a host and immune response with pro- and anti-inflammatory pathway) to infection (sepsis = infection + systemic inflammation response + organ dysfunction) [1]. In this new definition, severe sepsis was no longer included as a category, as all sepsis was, by its very nature, deemed to be severe.

The Surviving Sepsis Campaign (SSC) first proposed guidelines for the management of sepsis in 2004 [8], which have since been revised regularly, most recently in October 2021 [5]. These guidelines focus on the rapid diagnosis of sepsis that will initiate immediate and appropriate treatment, and include recommendations for the: (1) timely administration of antimicrobial therapy; (2) implementation of source control measures; (3) prevention of septic shock progression; and (4) reduction of sepsis-related morbidity and mortality [5]. The current sepsis definitions for children were proposed in 2005 and, while otherwise like those proposed for adults, included the 1992 SIRS criteria [9]. The latest SSC management recommendations for sepsis in children were published in 2020.

## 2. Review Objectives

The aim of this review of current international recommendations is to analyze the place of biomarkers in the diagnosis of sepsis, infection, systemic inflammation response and organ dysfunction, and to discuss the results, with a view to determining what measures might be proposed to ensure the best use and study of biomarkers as a tool in the future. Our aim is not to review and compare all existing and promising biomarkers, and we instead seek to define a standard and homogeneous framework for comparing them.

## 3. Review of Current Recommendations

In adults and children, the diagnosis of sepsis is based on an early clinical suspicion of infection, and then the identification of a dysregulated inflammatory response of the body that results in organ dysfunction [5,10]; both must be employed concurrently to determine therapeutic next steps. Therapeutic strategies that address the infection implicated in sepsis include the source control of infection (with surgery or drainage) and treatment with antimicrobials. Therapeutic approaches applied to address the dysregulated response of the body (systemic inflammation response) and organ dysfunction, might include adjunctive therapies (e.g., glucocorticoids) that are introduced alongside supportive care, including volume resuscitation and organ support (e.g., mechanical ventilation and renal replacement therapy).

### 3.1. Diagnosis of Suspected Infection

In the case of suspected sepsis, it is recommended to confirm or rule out bacterial infection and to continuously reassess, with a view to the initiation, modification, or discontinuation of antimicrobial therapy [5]. Unfortunately bacterial cultures, which are considered the gold standard for diagnosing bacterial infections, typically require 10 to 24 h to yield etiologic results [11], and an additional 8 to 24 h to yield antibiotic susceptibility results [12]. Although clinicians may already adjust antimicrobial treatment in about 20% of cases on the basis of the Gram results of a positive blood culture, and in an additional ~35% of cases on the basis of a matrix-assisted laser desorption-ionisation-time-of-flight (MALDI-TOF) identification obtained from a positive blood culture pellet [13], rapid antimicrobial susceptibility results are still warranted in nearly 50% of subjects with Gram negative bacteraemia. Even though faster technologies that provide antibiotic susceptibility results in 3–7 h have been developed to drive the tailoring of empiric antibiotic treatment, their turnaround time is insufficiently rapid to impact initial empirical therapy [13]. Furthermore, bacterial cultures can yield false negative results, potentially because of prior antibiotic therapy (high sensitivity of *Streptococcus* and *Neisseria* species) or inadequate sampling/an improper technique (air in anaerobic bottles may prevent the growth of most strict anaerobes).

It is proposed that antibiotic therapy should be initiated as soon as possible (within 1 h of recognition) in cases where there is a suspected septic shock (*SSC strong recommendation*, *low quality of evidence*) or a high likelihood of sepsis (*SSC very low quality of evidence*) [5]. Initiating broad-spectrum antimicrobial therapy in all patients with suspected infection solely on the basis of rapid clinical assessment (including history, clinical examination and tests that evaluate both infectious and non-infectious causes of acute illness) (*SSC Best Practice Statement*) [5], will result in unnecessary treatments in 60% to 70% of patients who ultimately do not have sepsis [14]. This mismatch occurs partly because the clinical presentation of viral and bacterial infections, as well as severe inflammatory processes, can be very similar. Such over-usage of antibiotics is problematic since: 1) antibiotics are associated with secondary toxic effects; and 2) contribute to the development of antimicrobial resistance (AMR), a significant concern in clinical practice. Overtreatment with antibiotics was particularly problematic during the COVID-19 pandemic [15,16], and it has more recently been estimated that, more generally, 4.95 million (3.62–6.57) worldwide deaths per year are associated with AMR [17].

Biomarkers of infection, such as procalcitonin (PCT), have been extensively studied to facilitate the rapid diagnosis of bacterial infection in patients with suspected sepsis. These biomarkers typically do not significantly enhance the sensitivity of infection diagnosis, due to the already high sensitivity achieved through scoring systems based on clinical signs and symptoms. Consequently, PCT is not recommended to aid decisions on when to start antimicrobials (*SSC weak recommendation*, *very low quality of evidence*) [5].

Ideally, biomarkers could play a role in improving specificity and restricting antimicrobial treatment to those patients with bacterial infection; a biomarker with a high negative predictive value (NPV) could also help rule out bacterial infection in patients with suspected sepsis and prevent unnecessary antibiotic treatments. We therefore believe that biomarkers, could serve as a valuable additional tool when used in this setting, and could guide clinicians on the decision “to give or not to give antibiotics?”.

Once antimicrobial therapy has been initiated, it should be reassessed daily for continuation, modification, or discontinuation, based on the results of bacterial cultures and clinical evolution (*SSC weak recommendation*, *very low quality of evidence*) [5]. Ultimately, the goal is to tailor antimicrobial therapy, both to each patient and to a given infection. A biomarker-based approach of this kind is aligned with the suggestion to privilege shorter (over longer) durations of antimicrobials (*SSC weak recommendation*, *very low quality of evidence*). For this purpose, the combination of PCT with clinical evaluation is suggested, as this will aid decisions on when to discontinue antimicrobials (*SSC weak recommendation*, *low quality of evidence*) [5]. A similar approach that uses C-reactive protein CRP has already been used but is not recommended in this instance [5,10].

### 3.2. Diagnosis of the Dysregulated Immune Response (including Systemic Inflammation) and Organ Failure

The diagnosis of the dysregulated immune response is based on the inflammatory response and organ dysfunction, according to the SSC 2021 guidelines [5]; and the diagnosis of systemic inflammation response is based on the systemic inflammatory response syndrome (SIRS) scoring system, rather than a biomarker value, such as CRP. The SIRS criteria for adults and children remain the same as in the (1992) original description [6], and incorporate three non-specific clinical parameters (temperature, heart rate, and respiratory rate), as well as laboratory testing for either circulating blood leukocytes or band forms.

As part of the clinical operationalization of the 2016 sepsis definition, the sequential organ failure assessment (SOFA) score [1], which assesses the function of six organ systems impacted by sepsis (lung, circulation/heart, brain, liver, kidney, coagulation), has been recommended for characterizing organ dysfunction. It is a relatively complex assessment that combines five clinical parameters and four laboratory results, and requires venous and arterial blood tests and approximately two hours for calculation [5]. In pediatric cases, the organ failure score is the pediatric logistic organ dysfunction (PELOD) score, which assesses the function of six organs [9].

The quick SOFA (qSOFA) score has been proposed as a sepsis screening tool, particularly in settings where all the parameters required to measure SOFA are unavailable. It is a simpler and quicker version that takes less than five minutes, and only measures three non-specific clinical parameters (blood pressure, mental status, and respiratory rate). Additional screening tools for organ dysfunction and illness severity include the national early warning score (NEWS) and the modified early warning score (MEWS): NEWS evaluates eight non-specific clinical parameters and three organs (lung, circulation/heart, and brain), and MEWS evaluates six non-specific clinical parameters and the same three organs. Neither of these scores requires blood sampling nor are they used in children. The 2021 SSC guidelines recommend discontinuing the use of qSOFA in favour of SIRS, NEWS, or MEWS, on the grounds of its poor sensitivity. (*SSC Strong recommendation*, *moderate quality of evidence*) [5].

Measuring serum lactate is suggested for adult patients suspected of having sepsis (*SSC Weak recommendation*, *low quality of evidence*) [1,5], and it can also be used as an additional assay to guide volume resuscitation in sepsis or septic shock cases (*SSC Weak recommendation; low quality of evidence*) [5]. However, it is neither sensitive nor specific enough to aid the early diagnosis of sepsis or dysregulated immune response and organ dysfunction. Lactate assay is often used in pediatrics but is not included in the 2020 pediatric guidelines [10].

### 3.3. Diagnosis of Sepsis

In short, the diagnosis of sepsis is based on the combined recognition of suspected infection + systemic inflammation response + organ dysfunction. At present, these three components are based on rapid clinical assessment that only uses non-specific tests and clinical scores (tools), not biomarkers (Table 1) [5]. 

### 3.4. Clinical Signs, Scores, and Biomarkers

With the evolving definition of sepsis and the absence of a reference test, clinicians are often confused about how to clinically diagnose sepsis. For example, in Switzerland, more than half of clinicians (53.8%) pragmatically use a mixture of the various sepsis-associated scores (e.g., SIRS, MEWS, NEWS, qSOFA, SOFA) [18]. Moreover, when sepsis is suspected, a total of 89.7% measure circulating blood leucocytes; 92.3% use CRP; 84.6% measure PCT; and 100% measure lactate. This variability suggests that clinicians might benefit from using accurate biomarkers rather than clinical scores in their daily practice.

In 2009, the International Sepsis Forum Colloquium on Biomarkers of Sepsis proposed to develop a systematic framework for the identification and validation of biomarkers of sepsis, and to promote collaboration between investigators, the biomarkers industry, and regulatory agencies [19]. Unfortunately, the numerous studies since carried out have remained rather heterogeneous and have not made it possible to identify and accurately compare one or more sepsis biomarkers with the capacity for diagnostic (including screening and monitoring), guiding therapeutic (theragnostic) or risk stratification/measuring surrogate endpoint (prognostic) purposes.

### 3.5. Biomarker and Sepsis Rapid Diagnosis (at Time of Clinical Suspicion)

Although most biomarker studies seek to show their utility in diagnosing suspected infection and sepsis, the use of biomarkers for rapid clinical assessment would ideally effectively and quickly (within 1 h) rule out the diagnosis of suspected infection and sepsis. The ideal diagnostic and theragnostic biomarker for rapid diagnosis of infection and sepsis should have a high negative predictive value (NPV) that minimizes false negatives for infection and sepsis (and therefore helps clinicians feel confident about not giving antibiotics after a negative test result) and a moderate positive predictive value (PPV) that minimizes false positives, and therefore helps confirm the clinical suspicion of sepsis.

### 3.6. Biomarker and Nosocomial Sepsis Screening (Pre-Symptomatic Diagnosis)

Ideally, diagnostic and theragnostic biomarkers should also be able to detect sepsis early, even before clinical suspicion arises. This early detection is only possible when there is routine screening (repetitive measurements) and would be most applicable in the pre-symptomatic diagnosis of hospitalized patients with a high risk of nosocomial sepsis. While screening is recommended for acutely ill and high-risk patients (*SSC Strong recommendation*, *moderate quality of evidence*) [5], the tools that could be used are not clearly specified, and could include variables analyzed by manual methods or the automated use of the electronic health record (EHR) analysis (with or without artificial intelligence). Variables that need to be considered include existing scores, vital signs, signs of infections, non-specific tests, and others (Table 1).

### 3.7. Biomarker and Monitoring

A diagnostic and theragnostic biomarker should be able to monitor the progression of sepsis and assist decisions related to antimicrobial continuation, modification, or discontinuation, which should be made in conjunction with clinical evaluation (*SSC Weak recommendation*, *very low quality of evidence*).

### 3.8. Biomarker to Stratify Risk and to Surrogate Endpoint

Finally, one or more prognostic biomarkers could also make a useful contribution to risk stratification and surrogate endpoints undertaken for triage and resource allocation purposes. In summary, when one or many more diagnostic, theragnostic and prognostic biomarkers are used in combination with clinical scores (to improve pre-test probability) or a machine learning system, this should contribute to the diagnosis, screening, monitoring, risk stratification and reporting of surrogate endpoints for sepsis or its components (i.e., infection, systemic inflammation response and organ dysfunction) (Table 1).

## 4. Discussion

Many biomarkers of sepsis have been studied. Prof. John Marshall and Prof. Jean-Louis Vincent, in their 2020 review that identified 258 different sepsis biomarkers [20], concluded: “Continuing to produce reports of novel biomarkers without developing a more rigorous framework to evaluate them and establish a recognized purpose is futile: it is time for a reappraisal of the possible roles of biomarkers in sepsis” [20]. From this point onwards, the conducting of novel robust prospective clinical studies that aim to better define the role of these different biomarkers for the diagnosis of infection, systemic inflammation response, and organ dysfunction should have been fundamental to sepsis biomarker evaluation, as such clinical studies of sepsis biomarkers could help to: (1) refine their usage and clinical scores; (2) analyze their performance characteristics (particularly PPV and NPV with cut-offs); and (3) specify their capacity to screen, diagnose, monitor, stratify risk, and demonstrate surrogate endpoints.

### 4.1. Ideal Criteria for a Sepsis Biomarker

We have previously proposed that a checklist should be established that outlines the basic requirements that sepsis biomarkers (infection, systemic inflammation response, and organ dysfunction) should meet [18]. Table 2 presents an updated and modified proposal of the ideal criteria for a sepsis biomarker for screening, diagnosing, and monitoring sepsis.

It will be necessary for the technology for determining the ideal sepsis biomarker(s) to be easily transferred from research laboratories to clinical use, and for it to meet regulatory criteria, such as the United States Food and Drug Administration 510(k) and the European In Vitro Diagnostic Regulation IVDR2022. In addition, the infection and sepsis biomarker results should be obtained within 1 h for patients with possible septic shock or a high likelihood of sepsis to quickly initiating or negating antimicrobial therapy (*SSC Strong recommendation*) [5,10]. A point-of-care test (POCT), with a <10 min dosing time, would be ideal for producing such a rapid result, and must be able to largely fulfill the ASSURE criteria (Affordable, Sensitive, Specific, User-friendly, Rapid, Equipment free) for a sepsis diagnosis test [21] (Table 2). In children, and especially in neonates, a capillary sample and less than 30–50 µL of blood volume would be ideal. Cost effectiveness studies should also be implemented to determine the financial consequences of sepsis biomarker testing for public health costs. Since the price of the biomarker assay is not expected to exceed 10 to 20 U$, and given the costs of a single additional intensive care unit (ICU) hospitalization day, we expect that the integration of biomarker testing into sepsis care pathways will not be cost adverse, and may even be cost beneficial.

### 4.2. Standardized Study Protocol for Sepsis Biomarker

Additionally, we propose a standardized protocol for diagnostic and theragnostic sepsis biomarker studies that will ensure comparability and address the pressing questions that surround the major public health issues of sepsis and AMR. This standardized protocol should include three specific sub-protocols, which each correspond to one of the three phases of sepsis (pre-sepsis = screening, sepsis = rapid diagnosis, post-sepsis = monitoring) that studies aim to investigate (Table 3).

The first, and very important, common points for all biomarker standard protocols are the definition of the infection and the definition of sepsis used. The yield from microbiologic testing can, however, be limited; for instance, the diagnosis of infection by bacteriological culture results is only possible in 30–40% of cases [14]. In this context, confirmation of infection diagnosis in a biomarker study must be reviewed by an adjudication committee (AC) independent of the investigators, who must reach their conclusions by referring to the analysis of several parameters and patient charts, and by applying the 2016 sepsis diagnostic criteria [5]. Taking into account two ways of diagnosing an infection is an approach that has been used in numerous studies comparing the treatment efficacy of different antibiotic regimens [22], and it has been proposed in different biomarker studies [23,24]. Firstly, the clinically documented infection (CDI), which occurs when clinical signs or symptoms and radiological or/and analytical findings indicate infection in the absence of microbiological proof, is considered. Then, the microbiologically documented infection (MDI), which includes bacteremia or a microbiologically documented local infection without positive blood cultures. Here it is important to note that only direct evidence of a micro-organism is considered sufficiently robust to consider an infection to be microbiologically documented, and this applies because indirect serology-based diagnosis may be falsely positive due to cross-reactive antibodies, and will therefore prove neither a current ongoing acute infection, nor where an infection of this kind has occurred.

In considering the biomarker evaluation of adult patients, it is therefore necessary to diagnose infection by using the aforementioned methods. In addition, classical biomarkers (e.g., CRP, PCT, circulating blood leukocytes, serum lactate) must be measured for each new biomarker determination, as well as the parameters used to calculate the SIRS-1992, NEWS, MEWS, SOFA or PELOD scores. However, the new biomarkers should not be compared with leukocytes, CRP, PCT and lactate, as these are not recommended biomarkers and are not considered to be gold standard. Biomarkers must therefore be compared with the adjudication committee’s diagnosis, based on the SSC 2021, CDI or MDI diagnostic criteria.

From a statistical perspective, it is necessary to analyze the specificity, sensitivity, receiver operating characteristic (ROC) curve, area under the ROC (AUROC) curve, PPV, and NPV, with cut-off values for both classical and new biomarkers. For sepsis screening, prioritizing high sensitivity will help ensure that false negative tests are minimized; in contrast, when rapid diagnosis at the time of clinical suspicion of sepsis is the priority, a test with high specificity that minimizes false positive tests is more important as it will ensure that the bedside clinician can be confident in a positive test result. Although AUROCs provide threshold values with the best combination of sensitivity and specificity, the interpretation of one biomarker should not however rely on a single threshold and on AUROC, in which surpassing it equates to a 100% risk of sepsis and falling below indicates 0% risk.

We instead propose defining four cut-offs and five levels of risk for each biomarker, and of categorizing the risk of infection, systemic inflammation response, organ dysfunction and finally sepsis as follows: very low at <10%, low at 10–20%, moderate at 20–50%, high at 50–80%, and very high at >80% (Table 4). This probability could then be integrated into a clinical score and incorporated into the clinician’s assessment. It should be noted that a single cut-off value combining the best biomarker sensitivity and specificity makes no clinical sense. On the other hand, a repetitive quantitative value and kinetics analysis may be proposed to exclude or confirm sepsis.

The goal of these observational, protocolized, and standardized studies is to pave the way for future interventional impact studies that can evaluate the real benefits of sepsis biomarkers, in terms of outcome measures (e.g., mortality, morbidity, length of hospitalization, frequency of recurrence/rehospitalization) and process of care endpoints [e.g., time to initiating antimicrobial treatment, other variables related to antimicrobial resistance (AMR); antimicrobial utilization (AMU) and antimicrobial consumption (AMC); results of complementary examinations (e.g., tests assessing for organ dysfunction)]. Ideally, a concurrent cost-effectiveness study should also be conducted to determine the cost of the biomarker assay and the savings made, and specific attention should be given to antibiotic, complementary examination, length of stay, and readmission rate costs, amongst others. 

Biomarker studies in the pre-sepsis phase (early screening of nosocomial sepsis) require more resources because all the common points described above must be analyzed every day or even several times a day. Statistical analyses are more complicated, especially for analyzing the kinetics with absolute and relative daily changes in biomarkers to predict infection and/or nosocomial sepsis. Biomarker studies in the sepsis phase (rapid diagnosis) require the determination of all the common parameters as soon as clinical sepsis is suspected. Time parameters, including the time between suspected sepsis, biomarker results, and antibiotic administration, must be considered and evaluated. Finally, a biomarker study of the post-sepsis phase (monitoring treatment efficacy) also requires frequent biomarker testing after the diagnosis of sepsis, as this is essential for determining treatment effectiveness, early detection of recurrence, and the duration of antibiotic therapy (including biomarker impact on de-escalation). Studies of this kind are more complex and require an AC to determine the adequacy of antibiotic treatment.

Examining combinations of different biomarkers could also improve the sensitivity and specificity, compared to considering a single biomarker in isolation. In engaging each biomarker, it might be interesting to develop a clinical pre-test probability score to further improve sensitivity and specificity. In addition, artificial intelligence/machine learning could potentially integrate all these parameters (biomarkers and clinical scores).

Classical biomarkers of (1) infection (blood leucocytes, PCT), (2) systemic inflammation response (CRP), and (3) organ failure (lactate), as well as new sepsis biomarkers [e.g., interleukin (IL)-6, presepsin, pancreatic stone protein (PSP), mid regional pro-adrenomedullin (MR-proADM), and others] should therefore be studied or re-studied by using the standardized approach proposed above, and this should be done in order to determine the biomarker(s) that address clinicians’ questions, and to identify the four cut-offs for the five level risks (Table 4). The aim of all these studies could be to improve knowledge of one or more infections, systemic inflammatory response, organ dysfunction and sepsis biomarkers, ensuring that all boxes on the checklist (Table 2) are ticked. This would help to address, and hopefully help to overcome, the major problems of sepsis clinical decision-making in the AMR era. 

## 5. Conclusions

This manuscript proposes a framework for conducting future clinically relevant translational biomarker studies for the diagnosis, prognosis, and management of sepsis, with the aim of identifying optimal indicators for infection, systemic inflammation response, organ dysfunction, and sepsis risk. These biomarkers should enable pre-symptomatic detection, guide antibiotic usage, stratify patients’ risk, and serve as surrogate endpoints. Both classical and new biomarkers should be evaluated to address specific clinical questions and establish risk categories. Rapid, point-of-care testing with prompt results that adheres to standardized criteria is essential, and economic considerations should also be considered, to ensure affordability.

Our findings and proposals may influence future biomarker studies and enable biomarkers to be proposed as tools by international guidelines, helping to address the critical challenges posed by sepsis in the AMR era.

## Figures and Tables

**Table 1 diagnostics-14-00300-t001:** Current Surviving Sepsis Campaign SSC 2021 recommendation to screen, diagnose, and monitor sepsis [5].

Surviving Sespsis Campaign SSC 2021	Early Screening In Acutely Ill and High-Risk Hospitalized Patients	Rapid Diagnosis At time of Clinical Suspicion	Monitoring Treatment Efficacy
**Suspected Infection**	*No recommendation*	Rapid clinical assessment with **Tests^b^** *Best Practice Statement* Against PCT to determine whether to initiate antimicrobial treatment *Weak recommendation* *Very low-quality of evidence*	Clinical evaluation with **Tests^b^** *Weak recommendation* *Very low-quality of evidence* For PCT (and clinical evaluation) to determine whether to discontinue antimicrobial treatment *Weak recommendation* *Low-quality of evidence*
**+** **S** **ystemic Inflam-** **mation response**	*No recommendation*	SIRS score *Strong recommendation* *Moderate quality of evidence*	*No recommendation*
**+Organ dysfunction**	*No recommendation*	NEWS, MEWS, SOFA *Strong recommendation* *Moderate quality of evidence* Lactate *Weak recommendation* *Low-quality of evidence*	Lactate Capillary refill time *Weak recommendation* *Low-quality of evidence* Dynamic parameters^c^ *Weak recommendation* *Very low-quality of evidence*
**=Sepsis**	**Tools^a^** *Strong recommendation* *Moderate quality of evidence*	Rapid clinical assessment with **Tests^b^** + NEWS, MEWS, SOFA lactate	Clinical evaluation with **Tests^b^** + PCT + Lactate + Capillary refill time + **Dynamic parameters^c^**

**Tools^a^**: Variables analysis by manual methods or automated use of the electronic health record (EHR) analysis (with or without artificial intelligence). Variables include scores, vital signs, signs of infections (unspecified tests), and others. **Tests^b^** (unspecified): biomarkers, microbiological investigations (serology, PCR, culture, etc.), radiological exams, other. **Dynamic parameters^c^**: response to passive leg raising or fluid bolus, stroke volume (SV), stroke volume variation (SVV), pulse pressure variation (PPV), or echocardiography. Abbreviation: systemic inflammation response syndrome (SIRS), sequential organ failure assessment (SOFA), quick SOFA (qSOFA), national early warning score (NEWS), modified early warning score (MEWS), procalcitonin (PCT).

**Table 2 diagnostics-14-00300-t002:** Modified checklist proposal—“Tick the box” Ideal criteria for a sepsis biomarker [18].

ASSURE Criteria for Ideal Sepsis Biomarker(s)	Early Screening In Acutely Ill and High-Risk Hospitalized Patient	Rapid Diagnosis At Time of Clinical Suspicion	Monitoring Treatment Efficacy
**Affordable**	☐	☐	☐
Price $10–20 US	☐	☐	☐
Cost-effectiveness study implemented	☐	☐	☐
**High Sensitivity**	☐		☐
Studies with a standardized protocol	☐		☐
**High Specificity**		☐	☐
Studies with a standardized protocol	☐	☐	☐
TP, TN, FP, FN percentages calculated	☐	☐	☐
**User-friendly testing**	☐	☐	☐
**Rapid**		☐	
Results within 1 h		☐	
Dosing time < 10 min		☐	
**Equipment free (or light)**	☐	☐	☐
POCT (price < $10,000 US)		☐	
**Pediatrics**	☐	☐	☐
Capillary blood	☐	☐	☐
Blood volume 30–50 μL	☐	☐	☐
**Certified**	☐	☐	☐
European IVDR	☐	☐	☐
FDA 510(k)	☐	☐	☐

Abbreviation: true positive (TP), true negative (TN), false positive (FP), false negative (FN); point of care testing (POCT); in-vitro diagnostic regulation (IVDR); United States Food and Drug Administration (FDA) 510(k) premarket notification.

**Table 3 diagnostics-14-00300-t003:** Standardized protocol for sepsis biomarker studies.

Study Design	Early Screening In Acutely Ill and High-Risk Hospitalized Patient	Rapid Diagnosis At Time of Clinical Suspicion	Monitoring Treatment Efficacy
**Data** ^a^	Daily recording Sepsis incidence Sepsis prevalence	Once Sepsis incidence Sepsis prevalence	Daily recording Sepsis incidence Sepsis prevalence
**Diagnostic** ^b^	Daily diagnosis	Once	Daily diagnosis
**Statistical analysis** ^c^	Daily Kinetics analysis	Once	Daily Kinetics analysis
**Endpoint/outcome** Diagnosis study Observational study	Accuracy of test for diagnosing infection, systemic inflammation response, organ dysfunction and sepsis	Accuracy of test for diagnosing infection, systemic inflammation response, organ dysfunction and sepsis	Accuracy of test for diagnosing infection and sepsis recovery

**Data**^a^: SIRS-1992, NEWS, MEWS, SOFA scores (or PELOD score for pediatrics), CRP, PCT, circulating blood leukocytes, serum lactate, and new biomarkers. **Diagnosis**^b^: by microbiological documented infection (MDI) or clinically documented infection (CDI) and adjudication committee (AC). **Statistical analysis**^c^: specificity, sensitivity, ROC curve, AUROC curve, PPV, and NPV, with cut-off values. Four cut-offs and five level of risk for each biomarker (Table 4). Combination of scores and/or biomarkers analyzing. Pre-test probability score. Abbreviation: systemic inflammatory response syndrome (SIRS); national early warning score (NEWS); modified early warning score (MEWS); sequential organ failure assessment (SOFA); pediatric logistic organ dysfunction (PELOD); C-reactive protein (CRP); procalcitonin (PCT); receiver operating characteristic (ROC); area under the ROC (AUROC), positive predictive value (PPV); negative predictive value (NPV).

**Table 4 diagnostics-14-00300-t004:** Four biomarkers’ cut-offs to assess five risk level of sepsis.

Risk of Sepsis	
Very High	>80%
High	50–80%
Moderate	20–50%
Low	<20%
Very Low	<10%

## Data Availability

Not applicable.

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
