# Peer review of "Proposed Framework for Conducting Clinically Relevant Translational Biomarker Research for the Diagnosis, Prognosis and Management of Sepsis"

_diagnostics, 2024, doi:10.3390/diagnostics14030300_

Round 1

Reviewer 1 Report

Comments and Suggestions for Authors

In this review, the authors criticise the sepsis guidelines that stand now.

Abstract: Although diagnosis of sepsis requires identification of the three components of infection, systemic inflammation response, and organ dysfunction, there are currently no consensus goldstandard criteria. There are, however, suggested tools and tests proposed in international guidelines, including those by the Surviving Sepsis Campaign. 

Sepsis-3 is the consensus definition of sepsis.

All of the "problems" stated in the manuscript have been recognised and all researchers studying sepsis are trying to find these cheap, reliable, sensitive, fast biomarkers.

So my main comment is, what is the point of this review? We all recognise the problems, we all know what a good biomarker should look like, what is your recommendation?

Author Response

Dear Reviewer #1,

Thank you for your review of our article.

Our intent was not to criticize current sepsis guidelines, such as those of the Surviving Sepsis Campaign SSC. In fact, one of the authors of our article, Shevin T. Jacob, is one of the authors of the SSC 2021 recommendations.

The Sepsis-3 definition, as you know, proposes the SOFA score as an operational tool.

Indeed, you are correct. Unfortunately, there is currently not an accepted gold-standard for the diagnosis of sepsis, and only clinical tools and scores are currently proposed.

We also know that biomarkers can be part of these tools and that it could therefore be interesting to find one or more biomarkers as future gold-standards.

The aim of this article is to help teams carry out well-constructed studies, not to highlight one biomarker or another, and specify the ideal criteria for such biomarkers. Indeed, back in 2009, several experts (Marshall JC, Reinhart K. Biomarkers of sepsis. Critical care medicine. 2009;37(7):2290-2298) and in 2020 (Pierrakos C, Velissaris D, Bisdorff M, Marshall JC, Vincent J-L. Biomarkers of sepsis: time for a reappraisal. Critical Care. 2020;24(1):1-15) said that future biomarker studies should be better designed.

Working in this field, we unfortunately still find that many studies on many biomarkers fail to answer useful clinical questions, and the aim of our article is to develop a standardized framework for conducting a valid biomarker study.

Our recommendation is that future biomarker studies should use a standardized protocol.

Conclusion

This manuscript outlines an approach to standardize sepsis biomarker research to identify optimal indicators for infection, systemic inflammation response, organ dysfunction, and sepsis risk. These biomarkers should enable pre-symptomatic detection, guide antibiotic usage, stratify patients’ risk, and serve as surrogate endpoints. Both classical and new biomarkers should be evaluated to address specific clinical questions and to establish risk categories. Rapid, point-of-care testing with prompt results is essential, adhering to standardized criteria. Economic considerations should ensure affordability. Our findings may influence future biomarkers studies and international guidelines, helping to address the critical challenges posed by sepsis in the era of AMR.

I hope our response will help you better understand our purpose, and please let us know if it doesn't, and what we should change in the article.

Best regards,

Dr François Ventura

Reviewer 2 Report

Comments and Suggestions for Authors

I read with interest this comprehensive commentary on sepsis biomarkers and the authors' subsequent proposal for a standardized research method in this field. The paper highlight all the all the limits and difficulties of the current definition of sepsis and the need for a systematic and shared approach in this field, first of all the absence of a reference gold standard. It should be noted, however, that the work appears more like an expert opinion rather than a real review. 

Therefore title may appear a little misleading. Biomarkers are generically cited as possible useful tools available to doctors for diagnosis, prognosis or for addressing therapy, but not systematically analyzed and compared with each other as one might have expected reading the title. 

Reviewing the current recommendations and with the exception of procalcitonin and PCR , which is quickly mentioned in the paragraph on the diagnosis of suspected infection,  the authors fails to indicates which biomarkers has been studied and could have promising clinical applications as markers of infection, dysregulated immune response or systemic inflammation. 

Table 1 with the recommendations of the SSC 2021 is very interesting, although not immediately readable, but it would be interesting to know if there are any biomarkers that can occupy the same boxes in the table and replace the indices proposed by the SSC 2021.

In the discussion the authors underline the need to clearly identify the use of the various biomarkers and study their performance in a comparable way, and I fully agree with this assumption. Nevertheless, as correctly highlighted by the authors in the text, the ideal characteristics of a biomarker depend on the intended use of it. it is unlikely that a single checklist will be sufficient to describe the ideal biomarker. The characteristics of cost, ease of execution, response time, statistical consistency (e.g. Sp, Se...), necessary equipment, depend on the function for which the biomarker was designed. Therefore it would be reasonable for Table 2 to contain at least three columns: early screening, rapid diagnosis and monitoring.

Some final suggestions: 

Table 3 lists in a note at the bottom of the table the data to be collected in a biomarker study. It would be useful to better highlight the minimum relevant data to be collected more clearly either with a separate table or with an explanation in the text. The acronyms MDI, CDI and AC are not explained.

In the conclusions paragraph it is written that " This manuscript outlines an approach to standardize sepsis biomarker research", I think it would be more correct to say that this manuscript "suggests a standardized approach".

Author Response

Dear Reviewer 2,

Thank you for your review of our manuscript.

You are correct. Our manuscript is an expert opinion based on a review of the current literature, which unfortunately shows sepsis biomarker studies fail to answer useful clinically relevant questions and/or gaps in knowledge. This problem has been appreciated since 2009 (Marshall JC, Reinhart K. Biomarkers of sepsis. Critical care medicine. 2009;37(7):2290-2298), but has yet to be resolved.

Yes, biomarkers are potential tools among others, and by choice we are not doing a complete review of all existing biomarkers (many recent studies have done so: https://pubmed.ncbi.nlm.nih.gov/36592205/ and https://pubmed.ncbi.nlm.nih.gov/34991675/ ) but we are trying to propose how to do well-constructed studies in the future. Current literature reviews on biomarkers do not allow any conclusions to be drawn, precisely because of the lack of standardization of biomarker studies.

We'll improve the visibility of Table 1, redo Table 2 according to your suggestions, improve Table 3 and change the conclusion according to your comments.

Best regards,

Dr François Ventura

Reviewer 3 Report

Comments and Suggestions for Authors

Dear Sirs, the authors have chosen a very uptodate issue to write for. Although the content is interesting and well-written, the authors should include and elaborate upon novel biomarkers that could be used for the diagnosis and therapeutics of sepsis. Therefore, in my opinion, the manuscript should be enriched by the meticulous description of these potential biomarkers and their prons and cons. In addition, the reference's section is too short, less than 30 references are too few for a review article.  From my point of view, the manuscript could be further proceed to publication after these major revisions.

Author Response

Dear Reviewer #3,

Thank you for your review of our manuscript.

The aim of this article is to help research teams carry out well-constructed studies, not to highlight one biomarker or another, and specify the ideal criteria for such biomarkers. Indeed, back in 2009, several experts (Marshall JC, Reinhart K. Biomarkers of sepsis. Critical care medicine. 2009;37(7):2290-2298) said that future biomarker studies should be better designed. In 2020, unfortunately, this is still not the case (Pierrakos C, Velissaris D, Bisdorff M, Marshall JC, Vincent J-L. Biomarkers of sepsis: time for a reappraisal. Critical Care. 2020;24(1):1-15)

Working in this field, we unfortunately still find that many studies on biomarkers fail to answer useful clinically relevant questions, and the aim of our article is to remind us (following on from that of 2009) and advise on how to conduct a meaningful and valid biomarker study.

Our recommendation is that future biomarker studies should use a standardized protocol.

There are already several reviews in the literature comparing various classical and novel biomarkers of sepsis (https://pubmed.ncbi.nlm.nih.gov/36592205/ and https://pubmed.ncbi.nlm.nih.gov/34991675/ ). Unfortunately, these reviews show just how heterogeneous study designs are, making it impossible to draw valid conclusions.

If all future publications follow our recommendations, it might be worthwhile to carry out a literature review of the various biomarkers.

Having intentionally chosen not to compare all biomarkers studied in sepsis, our article mentions only 30 references. Instead, our manuscript focuses on study design and development of a standardized framework for studies of sepsis biomarkers.

I hope you'll understand our choice and that you will be satisfied by the revised submission, with a few small changes suggested by the other reviewers.

Best regards,

Dr François Ventura

Round 2

Reviewer 1 Report

Comments and Suggestions for Authors

I thank the authors for adding an objectives section.

The review now makes more sense.

Author Response

Dear,

Thank you for your help in the final review of the article. We will change the title again at the request of another reviewer, and the objective of our review will probably be even clearer.

Best regards

François

Reviewer 2 Report

Comments and Suggestions for Authors

I appreciated the authors' effort to correct their article according to my observations, however I believe that the title of the review remains misleading. A sentence like "time for a standardized and systematic approach in future studies", should be added to the title.

I have  no other comments

Author Response

Dear,

Thank you for your idea of changing the title. It will be clearer this way.

Here's our proposal for a new title.

Time for a standardized and systemic approach in future studies to identify ideal biomarkers to screen, diagnose and monitor sepsis. 

Best regards

François

Reviewer 3 Report

Comments and Suggestions for Authors

cDear Sirs, the authors have not addressed the issues that were requested. Therefore, I recommend not accepting this manuscript for publication, as it does not offer significant information to the readers.

Author Response

Dear,

Our aim is not to review all 258 known and studied biomarkers of sepsis. In addition to the "Review objectives" paragraph, we're going to change the title at the request of another reviewer to make it even clearer. Once future studies have been standardized, it will be possible to compare the various biomarkers precisely on the basis of this systematic approach. 

The new title proposal is:

Time for a standardized and systemic approach in future studies to identify ideal biomarkers to screen, diagnose and monitor sepsis. 

For your information, we have just published a meticulous review of the literature on a new sepsis biomarker Pancreatic Stone Protein PSP. https://esmed.org/MRA/mra/article/view/4893   Best regards,   François

Round 3

Reviewer 3 Report

Comments and Suggestions for Authors

Accept in current form

Author Response

Thanks